# Evans Syndrome as a Possible Complication of Brentuximab Vedotin Therapy for Peripheral T Cell Lymphoma

**Ruxandra Irimia [1,2,\*], Sinziana Barbu [1,2], Codruta Popa [1,2] and Sorina Badelita [2]**

[1] School of Medicine, "Carol Davila" University of Medicine and Pharmacy, 4192910 Bucharest, Romania; sinziana.baitan@gmail.com (S.B.); codrumaia@yahoo.com (C.P.)

[2] Fundeni Clinical Institute, 4192910 Bucharest, Romania; sorinabadelita@gmail.com

\* Correspondence: ruxandra-maria.irimia@drd.umfcd.ro; Tel.: +004-0748-15-28-80

**Abstract:** Recently, Brentuximab Vedotin (BV) has emerged as an important therapy not only for Hodgkin's Lymphoma, but also for CD30-positive T cell lymphomas. Although anemia and thrombocytopenia are common myelosuppressive side effects, to our knowledge, this is the first described case of Evans Syndrome associated with BV therapy. We present the case of a 64-year-old female, diagnosed with relapsed Peripheral T Cell Lymphoma Not Otherwise Specified (PTCL-NOS), who, after receiving six cycles of BV, developed authentic severe autoimmune hemolytic anemia with strong positive direct anti-globulin (Coombs) test, simultaneously associated with severe immune thrombocytopenia. The patient was unresponsive to systemic corticotherapy, but fully recovered after a course of IV immunoglobulin.

**Keywords:** Evans syndrome; autoimmune hemolytic anemia; autoimmune thrombocytopenia; Brentuximab Vedotin; peripheral T cell lymphoma





## 1. Introduction

First described in 1951, Evans syndrome is a rare autoimmune disease characterized by simultaneous or sequential Coombs-positive auto-immune hemolytic anemia (AIHA) associated with immune thrombocytopenia and/or immune neutropenia [1–3]. Although the etiology is largely unknown, and in most cases is regarded as a primary/idiopathic condition, ES may develop as a secondary syndrome in adult patients with underlying conditions such as systemic lupus erythematosus, lymphoproliferative disorders, primary immunodeficiencies or bone marrow transplant. Since the outcomes of patients developing secondary ES are worse compared to primary ES, a comprehensive workup is necessary to distinguish between the two [4,5].

From a pathophysiological point of view, ES appears to be the result of a complex of immune dysregulations, rather than just an incidental association of immune cytopenias.

Although the exact pathophysiological mechanism is still poorly defined, the existing data suggest that the cytopenias characteristic for ES are related to T-cell abnormalities, including chronic T cell activation, reduced levels of helper T cells and increased proportions of cytotoxic T cells, similar to the alterations found in congenital hypoplastic anemia and amegakaryocytic thrombocytopenia [2,6].

The accurate ES diagnosis requires the exclusion of other causes of acquired immune cytopenias such as acquired immunodeficiency syndrome (AIDS); autoimmune lymphoproliferative syndrome (ALPS); or conditions that associate anemia and thrombocytopenia such as paroxysmal nocturnal hemoglobinuria (PNH), thrombotic thrombocytopenia purpura (TTP) or hemolytic uremic syndrome (HUS) [1,7,8].

Most of the data available on Evans syndrome come from the pediatric population, while the characteristics and outcome of adult patients are still to be defined. The management of Evans syndrome is challenging due to the rarity of the disease, but options for the first line of therapy include corticosteroids and intravenous immunoglobulin (IVIg), while

second line options are immunosuppressive agents, the monoclonal antibody rituximab, chemotherapy, danazol or a combination of these agents [1,7].

Brentuximab Vedotin (BV) is an antibody–drug conjugate directed against CD30 antigen.

It consists of the chimeric monoclonal antibody brentuximab which targets the cell-membrane protein CD30, linked to the antimitotic agent monomethyl auristatin E [8].

BV has been FDA and EMEA approved for previously untreated or relapsed systemic anaplastic large cell lymphoma or other CD30-expressing peripheral T-cell lymphomas, including angioimmunoblastic T-cell lymphoma and PTCLNOS [9–13].

The most frequent side effects include peripheral neuropathy, nausea, fatigue, diarrhea and pyrexia, while the myelosuppressive effects include neutropenia, anemia and thrombocytopenia.

To our knowledge, this is the first described case of Brentuximab Vedotin-induced Evans syndrome in a patient with PTCLNOS.

## 2. Case Description

We present the case of a 64-year-old woman, with a history of type II diabetes, that was first diagnosed in our center in 2009 with PTCL-NOS. She received four courses of CHOP (cyclophosphamide, doxorubicin, vincristine, dexamethasone) chemotherapy with no response, followed by DHAP (dexamethasone, high dose cytarabine, cisplatin), right cervical radiotherapy and autologous stem cell transplantation.

In January 2020, she developed a 5 cm diameter region of erythematous, indurated tegument and subcutaneous tissues on the left cervical region. The regional ultrasound performed in another medical facility showed densification of the subcutaneous tissues, with a width of approximately 6 mm. A skin biopsy was performed, and the histopathologic and immunohistochemical exam was consistent with a diagnosis of PTCL-NOS (CD4+/CD8−, CD3+, CD5+, CD7−, CD30+ heterogeneously, ALK-).

In March 2020, she was admitted in our department. The clinical exam revealed a 6 cm diameter left cervical erythematous induration of the latero-cervical area. The ultrasound showed a densification of the subcutaneous tissues without associated lymphadenopathies.

A CT of the cervical, thoracic, abdominal and pelvic regions was performed without any pathological findings. The FDG-PET/CT could not be performed due to reimbursement issues. The blood count showed no abnormalities, while the LDH level was within the normal range.

In April 2020, the erythematous skin lesions extended towards the anterior thoracic wall with associated induration/infiltration of the subcutaneous tissue stretching towards the left supraclavicular region. The clinical exam also revealed newly developed lymphadenopathy in the latero-cervical and supraclavicular region with a maximum diameter of 1.5 cm.

The blood count was consistent with mild anemia (Hb 10.9 g/dL), with a normal white blood cells and platelet count. The bone marrow smear was unremarkable, with 4–6% blastic cells and 5% mature lymphocytes.

We decided to initiate Brentuximab therapy (1.8 mg/kg) with appropriate premedication. Following the first infusion, the patient developed grade III infusion-reaction, successfully managed.

After the first administration, the peripheral lymphadenopathies disappeared and the skin lesions aspect was improved; full remission followed after three courses of BV.

The patient developed grade II peripheral neuropathy after the fourth cycle of Brentuximab, for which we initiated neurotrophic agents with significant symptom improvement.

On 7 August, the patient received the sixth course of Brentuximab. The clinical exam showed no abnormalities, and the complete blood count was in the normal range.

On 22 August 2020, the patient was admitted with an altered state, intense pallor, purpuric lesions on the lower limbs, petechiae and hemorrhagic bullae on the oral mucosa.

The blood count showed severe anemia (Hb 6.5 g/dL), reticulocytosis (7.2%), severe thrombocytopenia (PLT 4000/μL); and a normal white blood cell count (WBC 4490/μL; Neu 1650/μL). The peripheral blood smear showed polychromasia, spherocytes and schistocytes. Laboratory investigation included classical biochemical parameters for evidence of hemolysis. The Total and Direct Bilirubin levels were increased (5.7 mg/dL and 5.1 mg/dL, respectively). The direct Coombs test was intensely positive for IgG, while the serum haptoglobin level was significantly decreased. Bone marrow examination was carried out to rule out other causes of bi-cytopenia.

The bone marrow smear showed increased ratio of red blood cell precursors (33–35%) and polymorphic megakaryocytes with active thrombopoiesis.

The bone marrow biopsy displayed erythrocyte precursors and megakaryocytes hyperplasia.

The immunophenotypic exam, performed from peripheral blood, described a percentage of 72.6% mature T cells with a decreased CD4 helper/CD8 suppressor ratio of 1.4%.

Serology for Cytomegalovirus, Parvovirus, Epstein Barr, HIV, hepatitis B and C virus infection, as well as tests for anti-nuclear, anti-double stranded DNA antibodies, Rheumatoid factor and anti-Citrullinated protein antibodies were negative, ruling out an underlying disease.

The diagnosis of Brentuximab Vedotin-induced secondary Evans syndrome was established based on physical examination and detailed laboratory features, after thorough exclusion of other acquired causes of immune cytopenia.

We initiated corticotherapy with Dexamethasone 20 mg days 1–4 (dose adjusted due to the association of type II diabetes), followed by subsequent decrease to 16 mg for another 3 days, in response to severe hyperglycemia, but without therapeutical effect. On 28 August, we administered IV immunoglobins (1 g/kg) followed by 1000 mg Cyclophosphamide on 29 August 2020, with a subsequent increase in the platelet count and hemoglobin level on 30 August (67.000/mmc and 10.3 g/dL, respectively).

The patient was discharged on 1 September and was started on a daily dose of Prednisone, slowly tapered over the course of the following weeks.

The blood count levels remained within the normal range and no further clinical or biochemical abnormalities were found on the subsequent evaluations, while BV administration ceased.

## 3. Discussion

Although Evans syndrome was initially characterized as a primary/idiopathic disease, it is essential for the patient outcome to establish the possible association with an underlying condition. So far, very few cases of drug-induced ES have been described, and to our knowledge, this is the first case describing a possible association between the use of BV and ES.

Even though few cases of T cell lymphoma associated with ES have been described, the existing data suggest the incidence is extremely low, with studies reporting a frequency of only 0.43% and a standardized incidence of 0.007/100.000 for the association [13,14]. In addition, although the patient relapsed over the course of the following year, there were no additional alterations consistent with the diagnosis of ES.

Moreover, the patient developed immune anemia and thrombocytopenia only after the initiation of sixth cycle of the drug, with blood count normalizing after the interruption of BV therapy. The immunophenotypic exam performed when ES was diagnosed showed a decreased CD4 helper/CD8 suppressor ratio, an immune dysregulation often described for other ES patients. Although there is no definitive way to source the exact cause of ES in this patient, we argue that if the immune dysregulation determined by the malignant T cell lymphoma would have been the primary cause of ES in this patient, the bi-cytopenia would have also developed earlier in the course of the disease or during the subsequent relapse.

To date, there are few reports about the exact effect of BV on the immune subsets in patients with T cell lymphoma. The studies performed on patients with Hodgkin's disease showed that BV acts as an immune modulator by targeting CD30 regulatory T cells. BV induced the recovery of regulatory T-cells (CD4+ CD25+ CD127+) dysfunction and of myeloid immune-suppression, reduced production of neutrophil—derived immunosuppressive cytokines, but it is still unclear what is the effect in patients with T cell malignancies [15].

Following the extensive workup and the appropriate diagnosis of ES, we successfully managed this patient with intravenous immunoglobulin associated with Cyclophosphamide, despite the initial corticosteroid refractoriness.

**Author Contributions:** Writing—original draft preparation, R.I.; writing—review and editing, R.I. and S.B. (Sinziana Barbu), C.P. and S.B. (Sorina Badelita). All authors have read and agreed to the published version of the manuscript.

**Funding:** This research received no external funding.

**Institutional Review Board Statement:** Not applicable.

**Informed Consent Statement:** Informed consent was obtained from the patient. Written informed consent has been obtained from the patient to publish this paper.

**Data Availability Statement:** No new data were created or analyzed in this study. Data sharing is not applicable in this article.

**Conflicts of Interest:** The authors declare no conflict of interest.

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
