# Peer review of "Evans Syndrome as a Possible Complication of Brentuximab Vedotin Therapy for Peripheral T Cell Lymphoma"

_hematolrep, doi:10.3390/hematolrep15010023_

Round 1

Reviewer 1 Report

·  Line 23 page 1 shorten Evans Syndorm as (ES)

·  Line 36: i think that is important to underline that nowadays the pathophysiological mechanism of ES remains unknown. Several pathophysiological mechanism have been proposed: deficiency of CTLA-4 (CD152) and LRBA, deficit of TPP2, decreased level of T helper and increased T cytotoxic cells with a low CD4:CD8 ratio.

·  Line 48: aslo consider for second line treatment azathioprine, ciclosporin, danazol,  mycophenolate  mofetil,  splenectomy

· Line 64: you can shorten with PTCL-NOS

· Lines 87 - 88: 4-6% of blastic cells or pathological lymphocites?

· Lines 116-119: please specify if serology for virus and test for autoimmune diseases were all negative

· Lines 123-125: please pecify better how many days of steroid for each dosage

· Desametasone was administered during therapy with Ig ev and cyclophosphamide or not?

· Line 156: I think that “what is the effect in patients with T cells malignancies” sounds better

Author Response

Good afternoon,

Thank you very much for your suggestions.

Please find my responses:

  • Line 23 page 1 shorten Evans Syndorm as (ES) - change was made
  • Line 36: i think that is important to underline that nowadays the pathophysiological mechanism of ES remains unknown. Several pathophysiological mechanism have been proposed: deficiency of CTLA-4 (CD152) and LRBA, deficit of TPP2, decreased level of T helper and increased T cytotoxic cells with a low CD4:CD8 ratio - change was made
  • Line 48: aslo consider for second line treatment azathioprine, ciclosporin, danazol,  mycophenolate  mofetil,  splenectomy - danazol added. the rest were included in the more broad group of immunosupressive or chimiotherapy due to space restrictions
  • Line 64: you can shorten with PTCL-NOS - change was made
  • Lines 87 - 88: 4-6% of blastic cells or pathological lymphocites? - normal blastic cells
  • Lines 116-119: please specify if serology for virus and test for autoimmune diseases were all negative - the are all negative. change was made
  • Lines 123-125: please specify better how many days of steroid for each dosage - change was made
  • Desametasone was administered during therapy with Ig ev and cyclophosphamide or not? - yes, dexamethasone was administered. change was made
  • Line 156: I think that “what is the effect in patients with T cells malignancies” sounds better 
  • - change was made

Reviewer 2 Report

Irimia et al. described a case of a patient with PTCL-NOS who presented Evans Syndrome as a complication of Brentuximab Vedotin therapy.

This reviewer has a minor comment.

First PTCL-NOS was diagnosed in 2009 and relapsed in January 2022 as a subcutaneous lesion. Do you have more information of immunophenotype of this PTCL-NOS (both at the time of diagnosis in 2009 and relapsed lesion in 2020)?

Authors may not have information of diagnostic sample in 2009, however, for the one in 2020, please add morphologic and immunophenotypic information to support the diagnosis of PTCL-NOS either in the main text or as a table, or as a figure (microscopic photos). Brentuximab was used, therefore I assume this lymphoma expressed CD30. Unfortunately, I cannot find the detailed histologic and immunophenotypic information.

Author Response

First PTCL-NOS was diagnosed in 2009 and relapsed in January 2022 as a subcutaneous lesion. Do you have more information of immunophenotype of this PTCL-NOS (both at the time of diagnosis in 2009 and relapsed lesion in 2020)? - we do not have the immunophenotype from the time of diagnosis in 2009

Authors may not have information of diagnostic sample in 2009, however, for the one in 2020, please add morphologic and immunophenotypic information to support the diagnosis of PTCL-NOS either in the main text or as a table, or as a figure (microscopic photos). Brentuximab was used, therefore I assume this lymphoma expressed CD30. Unfortunately, I cannot find the detailed histologic and immunophenotypic information. - change was made, additional information was added. The lymphoma was indeed CD30 positive

Reviewer 3 Report

Dear authors, 

Thank you for this case report. The introduction is complete and scientifically relevant, including recent and relevant scientific references. The case description also includes all the information necessary to understand the clinical case but needs to be improved on some minor points. The discussion needs to be largely adapted. Most of the work is done but some assertions cannot be defended medically.

Major point: 

Line 146-147:  Because the exact pathophysiologic mechanism of Evans syndrome is currently unknown, it is very difficult to differentiate in this patient between Evans syndrome secondary to BV or secondary to T lymphoma. According to the authors, the kinetics of onset of cytopenias differentiate between ES secondary to T lymphoma or ES secondary to BV (Line 146-147). Furthermore, the authors support their hypothesis by mentioning that the CD4/CD8 ratio is decreased at the time of diagnosis of ES but do not specify whether this ratio was already decreased before or not. Moreover, they also justify it by a normalization of the blood count after discontinuation of treatment, but the avoidance of treatment and normalization of the blood count cannot support a diagnosis of Evans syndrome secondary to BV since the kinetics and pathophysiology are not known (as it could be done for drug-induced cytopenias, for instance).   Although the diagnosis of Evans syndrome is definite, it is impossible to conclude with certainty that it is secondary to Brentuximab Vendotin and not to the T lymphoma. The causal link cannot be asserted, and this would risk misleading the reader. Therefore, this is a hypothesis and not an assertion. Please modify the all manuscript by being careful about the origin of the syndrome (Title, Abstract and discussion).

- Minor points : 

o Discussion: Authors should precise if there was a FDG-PET/CT performed and if not, why only a single CT. FDG-PET/CT is the current state-of-the-art imaging in lymphoma.

o All manuscript : Please use SI units.  Blood count units: "u" is stricto sensu not "µ". Line 102 : Precise "WBC" and "Neu" is not scientifically correct.

o Line 75 : “erythematous induration” should be separated.

o Line 88 : lymphocYtes.

Author Response

Good evening,

Thank you very much for your comments.

Please find in red my response:

Line 146-147:  Because the exact pathophysiologic mechanism of Evans syndrome is currently unknown, it is very difficult to differentiate in this patient between Evans syndrome secondary to BV or secondary to T lymphoma. According to the authors, the kinetics of onset of cytopenias differentiate between ES secondary to T lymphoma or ES secondary to BV (Line 146-147). Furthermore, the authors support their hypothesis by mentioning that the CD4/CD8 ratio is decreased at the time of diagnosis of ES but do not specify whether this ratio was already decreased before or not. Moreover, they also justify it by a normalization of the blood count after discontinuation of treatment, but the avoidance of treatment and normalization of the blood count cannot support a diagnosis of Evans syndrome secondary to BV since the kinetics and pathophysiology are not known (as it could be done for drug-induced cytopenias, for instance).   Although the diagnosis of Evans syndrome is definite, it is impossible to conclude with certainty that it is secondary to Brentuximab Vendotin and not to the T lymphoma. The causal link cannot be asserted, and this would risk misleading the reader. Therefore, this is a hypothesis and not an assertion. Please modify the all manuscript by being careful about the origin of the syndrome (Title, Abstract and discussion). - the patient relapsed since January 2022 and although we tried all the available treatment options the patient is not responding to any treatment. Despite the current state, the patient never developed any cytopenias that could be consistent with the presentation of Evans Syndrome. We believe that since the episode suggestive for Evans Syndrome was only seen once in the 13 year disease history of this patient, the association with BV therapy is very likely. 

on line 139 we added- this is the first case describing a possible association between the use of BV and ES 

line 144 we added - In addition, although the patient relapsed over the course of the following year, there were no additional alterations consistent with the diagnosis of ES.

line 149: Although there is no definitive way to source the exact cause of ES in this patient, we argue that if the immune dysregulation determined by the malignant T cell lymphoma would have been the primary cause of ES in this patient, the bi-cytopenia would have also developed earlier in the course of the disease or during the subsequent relapse. 

Title: Evans Syndrome as a possible complication of Brentuximab Vedotin therapy for Peripheral T Cell Lymphoma

- Minor points : 

o Discussion: Authors should precise if there was a FDG-PET/CT performed and if not, why only a single CT. FDG-PET/CT is the current state-of-the-art imaging in lymphoma. -FDG-PET/CT was not performed since in Romania it is reimbursed only in certain situations and the price was prohibitive for the patient. Information was added on line 79

o All manuscript : Please use SI units.  Blood count units: "u" is stricto sensu not "µ". Line 102 : Precise "WBC" and "Neu" is not scientifically correct. - change was made

o Line 75 : “erythematous induration” should be separated.- change was made

o Line 88 : lymphocYtes.- change was made

Round 2

Reviewer 3 Report

Dear authors, 

thank you for making the changes to the manuscript. The reader can now qualify the diagnosis of secondary Evans syndrome with the newly balanced evidence. Indeed, it is the kinetics of the early onset of cytopenias following the introduction of the drug that has argued for this diagnosis. The addition of the paragraph (line 149) is explicit about this.